## [Reviewer comments · BMJ Open]

ARTICLE DETAILS

TITLE (PROVISIONAL)	Protocol for the mWellcare Trial: A multi-centre, cluster randomized, 12-month, controlled trial to compare the effectiveness of mWellcare, an mHealth system for an integrated management of patients with hypertension and diabetes, versus enhanced-usual care in India
AUTHORS	Jha, Dilip; Gupta, Priti; Vamadevan, Ajay; Jindal, Devraj; Perel, Pablo; Prieto-Merino, David; Jacob, Pramod; Nyong, Jonathan; Venugopal, Vidya; Singh, Kavita; Goenka, Shifalika; Roy, Ambuj; Tandon, Nikhil; Patel, Vikram; Prabhakaran, Dorairaj

VERSION 1 - REVIEW

REVIEWER	David Peiris The George Institute for Global Health, University of Sydney, Australia I have co-published two papers with Professor Prabhakaran on a similar trial being conducted in Andhra Pradesh
REVIEW RETURNED	22-Dec-2016

GENERAL COMMENTS	Thank you for the opportunity to review this c-RCT protocol of a multifaceted intervention to improve BP and diabetes care in 40 community health centres in Haryana and Karnataka, India. The protocol is well-structured and easy to follow. I have a few clarifications listed below: 1. What is the current status if the trial? The verb tense shifts from past, present and future and it looks like a lot of things have already been implemented?2. Some more recent references on mHealth evidence which may be useful to include:• Hall AK, et al. Mobile Text Messaging for Health: A Systematic Review of Reviews. Annual Review of Public Health. 2015;36(1):393-415.• Free C, et al. The Effectiveness of Mobile-Health Technologies to Improve Health Care Service Delivery Processes: A Systematic Review and Meta-Analysis. PLoS Med. 2013;10(1):e1001363.• Peiris D, et al. Use of mHealth Systems and Tools for Non-Communicable Diseases in Low- and Middle-Income Countries: a Systematic Review. J of Cardiovasc Trans Res. 2014:1-15.3. Are there any inclusion/ exclusion criteria around access to a mobile phone to receive the SMS reminders. What level of phone ownership is needed – personal, shared etc. How are the investigators handling literacy as a barrier to engaging in SMS?
--

	4. Inclusion criteria are based on BP and blood glucose elevations regardless of past history? Later on it says that all individuals with diagnosed hypertension and diabetes will be assessed by the nurse. Are people with a known diagnosis whose BP and/or blood glucose are not elevated therefore excluded from participation? 5. The mERA checklist could be useful for describing the intervention in more detail. http://www.bmj.com/content/352/bmj.i1174 6. In Haryana, the authors state there are no nurses employed under the NPCDCS. Will the research team be employing these nurses to deliver the intervention and/or enhanced care? How many nurses and doctors are to be trained? Will this be in addition to their usual work or are they dedicated positions? How will they be remunerated? 7. Are patients recruited first and then CHCs randomised or vice versa? It is difficult to ascertain from the flow chart at what point randomisation occurs. 8. It looks like independent evaluators are doing the outcome assessment, but it is not clear if the nurse doing the data collection at baseline is independent of the CHC staff or one and the same. Are they blinded to treatment allocation (if randomization occurs before recruitment)? 9. Are any pre-specified sub-group analyses to be conducted (e.g. region, diabetes and hypertension groups etc.)? 10. I am a little unclear of the role of the DSMB in this trial. Given it is only 12 months' duration and data appear to be only collected at baseline and end of 12 months what will the DSMB actually do?
--	--

VERSION 1 – AUTHOR RESPONSE

Response to Reviewer's comments:

1. We have looked at the verb tense carefully and made the necessary changes to reflect the current status. Currently the recruitment is complete and the intervention are ongoing.
2. Thank you for these suggestions. We have included the references now. Please see page 5 & 13 of revised manuscript.
3. Ownership or access to mobile phone is not a part of inclusion/exclusion criteria for trial participation. All eligible participants irrespective of mobile phone ownership or access were invited to participate in the trial. At the time of recruitment participants were asked to provide their preferred mobile number if they opted for receiving SMS reminder. Regarding literacy issue, the trial steering committee discussed the possibility of using interactive voice response (IVR) instead of SMS. However, the committee favoured SMS as it can be received anytime and accessed as per participant's convenience whereas the participant should personally the IVR call to listen to the message. Illiterate participants were given the option of obtaining help from their immediate relatives or friends if needed and those who agreed were provided SMS.
4. Yes, those individuals whose blood pressure or fasting plasma glucose were controlled i.e. blood pressure <140/90 mmHg or fasting plasma glucose <140 mg/dl or post prandial glucose <200 mg/dl were not included in the trial. However, to maintain the uniformity of care provided at the CHCs, all individuals with diagnosed hypertension and diabetes were provided care using mWellcare system in the intervention arm.

5. Thank you for the suggestion. Since the focus of this paper is on describing the trial methodology, we have provided limited details on technology platform, infrastructure, interoperability etc. However, we are preparing another paper providing details of the mHealth intervention and process of its development where we will refer to the mERA checklist. Also we will refer to the checklist at the time of reporting trial results.
6. In Haryana, the research team has employed nurses for all 20 trial sites (intervention and enhanced care). In addition to providing intervention/enhanced care, these nurses are providing care as per the NPCDCS program to all patients attending the NCD clinic. Nurses are being remunerated as per the Government of Haryana norms from the mWellcare project funds. 40 Nurses and 40 doctors were trained for the trial. In addition, on site orientation was regarding the trial was provided to all doctors at the intervention sites to cover the changes in duty roster.
7. CHCs were randomized before participant recruitment. We have modified Figure-1 and added randomization before participant recruitment.
8. Nurse conducting the baseline data collection is a CHC staff. Besides baseline data collection, she is responsible for intervention/enhanced care delivery and other duties assigned under NPCDCS. Nurses are not blinded to treatment allocation.
9. In the protocol we specified that we would repeat the main analysis by state (as well as for the whole sample). Sample size was calculated to have enough power for this. Some outcomes are only relevant and measured in some individuals, such as glycemia in patients with diabetes. We also had a sentence in the protocol stating that we would do subgroup analysis but at the time we did not specify any particular grouping variables. However in the Statistical Analysis Plan we will specify the subgroup analysis.
10. DSMB was constituted to review trial protocol, monitor data safety and protect safety of study participants. Taking into account short duration of the study, the DSMB will be meeting every 6 months. We have done this as a matter of abundant caution particularly not to miss the likelihood of a huge benefit.